# Specific DNMT3C flanking sequence preferences facilitate methylation of young murine retrotransposons
Leonie Dossmann[1], Max Emperle[1], Michael Dukatz[1], Alex de Mendoza [2], Pavel Bashtrykov[1] & Albert Jeltsch [1]✉

The DNA methyltransferase DNMT3C appeared as a duplication of the *DNMT3B* gene in muroids and is required for silencing of young retrotransposons in the male germline. Using specialized assay systems, we investigate the flanking sequence preferences of DNMT3C and observe characteristic preferences for cytosine at the -2 and -1 flank that are unique among DNMT3 enzymes. We identify two amino acids in the catalytic domain of DNMT3C (C543 and V547) that are responsible for the DNMT3C-specific flanking sequence preferences and evolutionary conserved in muroids. Reanalysis of published data shows that DNMT3C flanking preferences are consistent with genome-wide methylation patterns in mouse ES cells only expressing DNMT3C. Strikingly, we show that CpG sites with the preferred flanking sequences of DNMT3C are enriched in murine retrotransposons that were previously identified as DNMT3C targets. Finally, we demonstrate experimentally that DNMT3C has elevated methylation activity on substrates derived from these biological targets. Our data show that DNMT3C flanking sequence preferences match the sequences of young murine retrotransposons which facilitates their methylation. By this, our data provide mechanistic insights into the molecular co-evolution of repeat elements and (epi)genetic defense systems dedicated to maintain genomic stability in mammals.

DNA methylation plays an essential regulatory role in chromatin regulation, including control of gene expression and silencing of transposable elements[1–5]. Moreover, it is involved in classical epigenetic processes like X-chromosome inactivation and parental imprinting. Genome-wide DNA methylation is essential for genome stability and necessary for cell differentiation. In vertebrates, DNA methylation occurs primarily at the C5 position of cytosine residues within CpG sites. While these CpG sites are depleted in the bulk genome, they are enriched in CpG islands, which are mostly found in promoter regions of genes and transposable elements[6]. De novo DNA methyltransferases establish DNA methylation patterns during gametogenesis and early embryonic development[3,7]. The genomes of most mammals contain two de novo DNA methyltransferases, DNMT3A and DNMT3B[8,9], and the DNMT3-like protein DNMT3L[10], which is catalytically inactive, but has regulatory functions as a cofactor in this process[7,11]. In mice and other muroids, a third DNMT3 protein, called DNMT3C, has more recently been discovered[12,13]. It is expressed in the male germline in prospermatogonia and spermatocytes[12], roughly overlapping with the

expression profile of DNMT3A2[14]. Nevertheless, DNMT3C is required for methylation and silencing of evolutionary young retroelements[12,13]. Later, methylation of the promoters of young mouse retrotransposons has been shown to depend on DNMT3C, but not on DNMT3A[15], and global DNA methylation analyses indicated that DNMT3C has a very specific role in the global DNA methylation, while DNMT3A acts more globally, and both activities are required for spermatogenesis[15]. Phylogenetic analyses revealed that DNMT3C has evolutionary developed from DNMT3B by gene duplication in the last common ancestor of muroid rodents[16]. DNMT3C and DNMT3B share high amino acid sequence similarity in their C-terminal parts, which contain the methyltransferase domains (Supplementary Fig. 1). The N-terminus shows lower sequence conservation, signs of positive selection[16], and DNMT3C lacks the PWWP domain, which is involved in chromatin targeting of DNMT3A and DNMT3B[7,11] (Fig. 1a).

Structural studies have shown that the C-terminal catalytic domains of DNMT3A and DNMT3B form a linear heterotetrameric complex with the

---

[1]Institute of Biochemistry and Technical Biochemistry, Department of Biochemistry, University of Stuttgart, Allmandring 31, 70569 Stuttgart, Germany. [2]School of Biological and Behavioural Sciences, Queen Mary University of London, Mile End Road, E1 4NS, London, UK. ✉e-mail: albert.jeltsch@ibtb.uni-stuttgart.de

**Fig. 1 | Structure and purification of DNMT3C catalytic domain proteins. a** Domain composition of different DNMT3 enzymes. The murine DNMT3C protein (Uniprot entry P0DOY1) consists of 740 amino acids. See also Supplementary Fig. 1. **b** Structure of the heterotetrameric complex of DNMT3B catalytic domain and DNMT3L C-terminal domain[19]. DNMT3B is shown in orange and brown, DNMT3L in grey. Residues differing in DNMT3C and DNMT3B are colored pink in the orange DNMT3B subunit. DNA is shown in blue with the -1 flank base pair in cyan and the -2 flank base pair in yellow. **c** Purification of WT DNMT3C catalytic domain and mutants. Lane 1: WT DNMT3C, lane 2: C543N/V547A mutant, lane 3: C543N mutant, lane 4: V547A mutant, lane 5: E590K mutant. M: Protein size marker. **d** Zoom into the DNMT3B/3L structure focusing on the -2 and -1 flank region. Coloring is as described in panel (**b**). N656 (corresponding to C543 in DNMT3C), A660 (V547 in DNMT3C) and K703 (E590 in DNMT3C) are highlighted. There are no other amino acids differences between DNMT3C and DNMT3B near the DNA at the -2/-1 flank region.

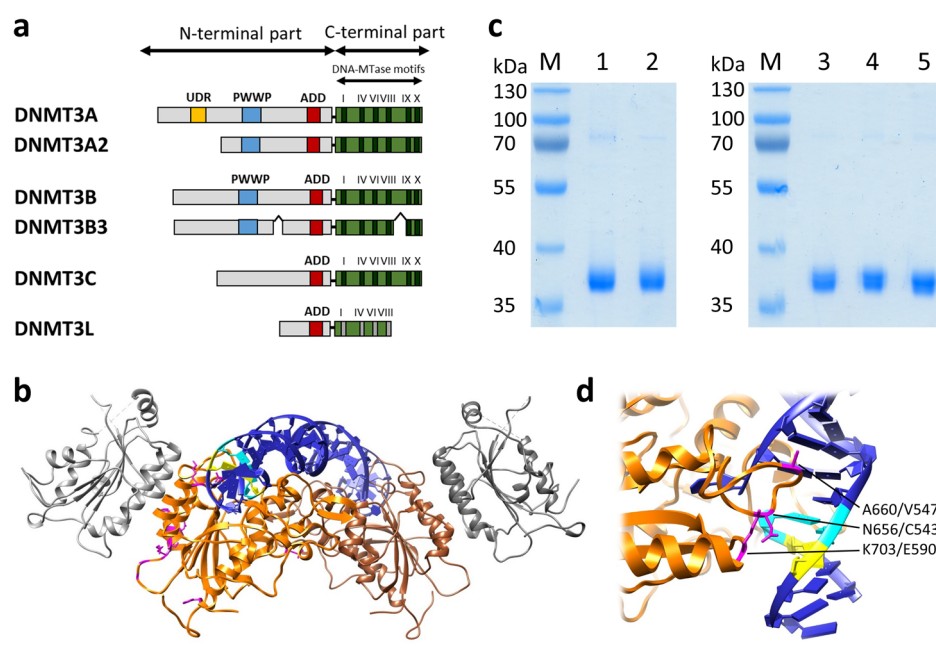

C-terminal domain of DNMT3L, where DNMT3A or DNMT3B subunits are positioned in the center and DNMT3L at the edges (Fig. 1b)[17–19]. However, DNMT3A and DNMT3B are also active in absence of DNMT3L and in this case additional DNMT3A/B subunits occupy places of the DNMT3L subunits forming homotetramers as smallest catalytically active unit, but larger homooligomers can assemble as well[20–22]. Based on the very high sequence similarity of the catalytic domains of DNMT3C and DNMT3B, DNMT3C is expected to adopt similar structures. Previous analyses revealed that DNMT3A is particularly involved in methylation of major satellite repeats and establishment of allele-specific imprinting during gametogenesis[23,24], while DNMT3B was found to methylate repetitive elements in centromeric minor DNA and retrotransposons like long interspersed nuclear element-1 (LINE-1) and human Satellite II (SatII) repeats[9,24–26]. Kinetic studies have shown that DNMT3 enzymes do not methylate all CpG sites equally, but their catalytic activity is modulated up to 100-fold by the adjacent base pairs called flanking sequence here[19,27–29]. These flanking sequence preferences differ strongly between DNMT3A and DNMT3B[19] and in a subtle manner also between human and mouse DNMT3B[28]. It has been shown that the divergence in flanking sequence preferences between DNMT3A and DNMT3B contributes to their differential activities on human SatII repeats, because CpG sites in a SatII flanking sequence context (ATTCGATG) are among the most preferred substrates of DNMT3B[19]. Moreover, the flanking sequence preferences of DNMT3A and DNMT3B correspond to the finding that DNMT3B methylates minor satellite repeats in mouse ES cells, whereas DNMT3A methylates major satellite repeats[24]. The subtle differences in the flanking sequence preferences of human and mouse DNMT3B also correlate with the sequences of mouse and human repeat sequences[28] suggesting a co-evolution of flanking sequence preferences and cellular DNMT targets in mammals. While DNMT3B is active during early embryogenesis[9], DNMT3C was found to be highly expressed in fetal stages of the male germline development, where it methylates promoter regions of evolutionarily young retrotransposons[12,13]. In the same developmental period, imprints and genomic methylation are introduced by DNMT3A together with DNMT3L[10,23,30]. In the absence of DNMT3C, spermatocytes are not able to complete their full development due to a failure in meiosis leading to apoptosis[12,13]. The reason for this defect was shown to be a lack of repression of evolutionary young retrotransposons, leading to activity of LINE-1 and ERVK elements, like IAPs[12,13].

While DNMT3A and DNMT3B have been studied for more than 20 years and many details regarding their structure, function and mechanism are resolved[7,11,31], it has remained unclear which specific biochemical or mechanistic features have led to the emergence of DNMT3C in muroids. As described above, previous work on DNMT3 enzymes has connected their flanking sequence preferences with their specific biological methylation targets. It is the aim of this study to identify the flanking preferences of DNMT3C and investigate, if they show adaptations towards the biological substrates of DNMT3C by comparison with the sequences from the promoter regions of their known target repeats. Interestingly, when comparing with DNMT3B, a clear shift in flanking sequence preference of DNMT3C at positions -2 and -1 could be shown that is evolutionary conserved in muroids and unique among the DNMT3 enzyme family. On the basis of DNMT3B structures, two amino acid residues in DNMT3C (C543 and V547) were identified to cause this DNMT3C-specific DNA interaction providing important mechanistic insight into the molecular basis of flanking sequence preferences of DNMT3 enzymes and their flanking sequence-specific DNA interaction. Comparison of DNMT3C flanking sequence preferences with the sequences of repeats targeted by this enzyme showed that DNMT3C is highly optimized to methylate its natural target sequences, providing an interesting example of the co-evolution of repeat elements and the mammalian DNA methylation machinery.

## Results
### DNMT3C shows unique flanking sequence preferences
Previous data illustrated that flanking sequence preferences of DNMT3B are adapted to its biological targets like satellite II repeats[19] and human vs. mouse repeats[28]. Given that DNMT3C appeared as gene duplication of DNMT3B in muroid rodents[16], where it was shown to methylate young retrotransposons in the male germline[12,13], we wanted to find out if its flanking sequence preferences also show particular adaptations to its biological targets. We therefore cloned, overexpressed and purified the catalytic domain of DNMT3C (Fig. 1c). The activity and flanking sequence preferences of DNMT3C were tested in Deep Enzymology experiments[32] using a library of DNA substrates, which contain a single CpG or CpN site in a sequence context of 10 randomized bases on either side (Fig. 2a). Afterwards, the analysis window was restricted to the -8 to +8 region to minimize the effects of the constant sequence outside of the randomized region on the derived profiles. Using an established workflow based on hairpin ligation, bisulfite conversion, NGS and bioinformatic analysis[19,28,29], we analyzed the

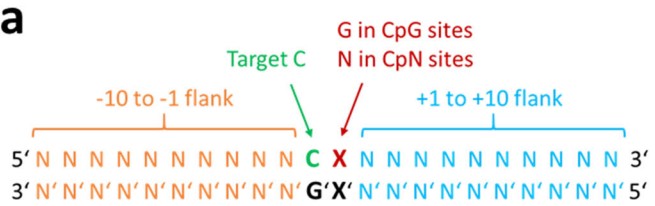

**Fig. 2 | Flanking sequence preference analysis of DNMT3 enzymes. a** Schematic picture of the design of the random flank substrates. **b-e** Combined -4 to +4 flanking profiles of WT DNMT3C, WT DNMT3B, WT DNMT3A and the DNMT3C C543N/V547A mutant. DNMT3A data were combined from[19,29]; DNMT3B data were from[19]. Shown are observed/expected (o/e) values for the occurrence of individual bases at each position in the methylated products. DNMT3C-specific C-preference at position -2 and -1 is highlighted by blue arrows, the DNMT3A and DNMT3B characteristic preference for T(-2) is labelled with a red arrow. See also Supplementary Figs. 2 and 3. **f** Correlation of the combined -8 to +8 flank profiles of the different DNMT3s. Shown are Pearson *r* values.

CpG specificity and flanking sequence preferences of DNMT3C. For a good coverage of the reaction progress curve, several independent methylation reactions were conducted using different enzyme concentrations and incubation times (Supplementary Table 1 and 2).

Starting with the analysis of the CpG methylation flanking preferences, the enrichment and depletion of individual bases in the -8 to +8 flanking region of methylated product molecules were extracted and expressed as observed/expected ratio (o/e). Despite different overall methylation levels of the individual data sets, all the methylation preferences of reactions were highly correlated (Supplementary Fig. 2). Therefore, they were merged to provide a first overview of the flanking sequence preferences of DNMT3C and compared it with the mouse DNMT3B profile taken from Gao et al. 2020[19] and DNMT3A profiles calculated with data taken from Gao et al.

2020[19] and Dukatz et al. 2022[29] (Fig. 2b-d, Supplementary Fig. 3a-c). The data revealed characteristic deviations in the flanking sequence preferences of DNMT3C and DNMT3B at the -2 and -1 flanking position. At the -2 position, DNMT3B (and DNMT3A) display a strong preference for T, but DNMT3C shows the highest preference for C(-2). At the -1 position, DNMT3B and DNMT3C both prefer A, but DNMT3C shows a strong gain in preference for C, resembling the DNMT3A pattern. Hence, DNMT3C differs from its closely related paralog DNMT3B by the increase in preference for C at the -2 and -1 positions. In contrast, at the +1 to +4 sites, DNMT3C shows DNMT3B-like flanking sequence preferences, which are clearly distinct from DNMT3A. In summary, the flanking sequence preferences of DNMT3C are more similar to DNMT3B than to DNMT3A (Fig. 2f), the C(-2) preference of DNMT3C is unique within the

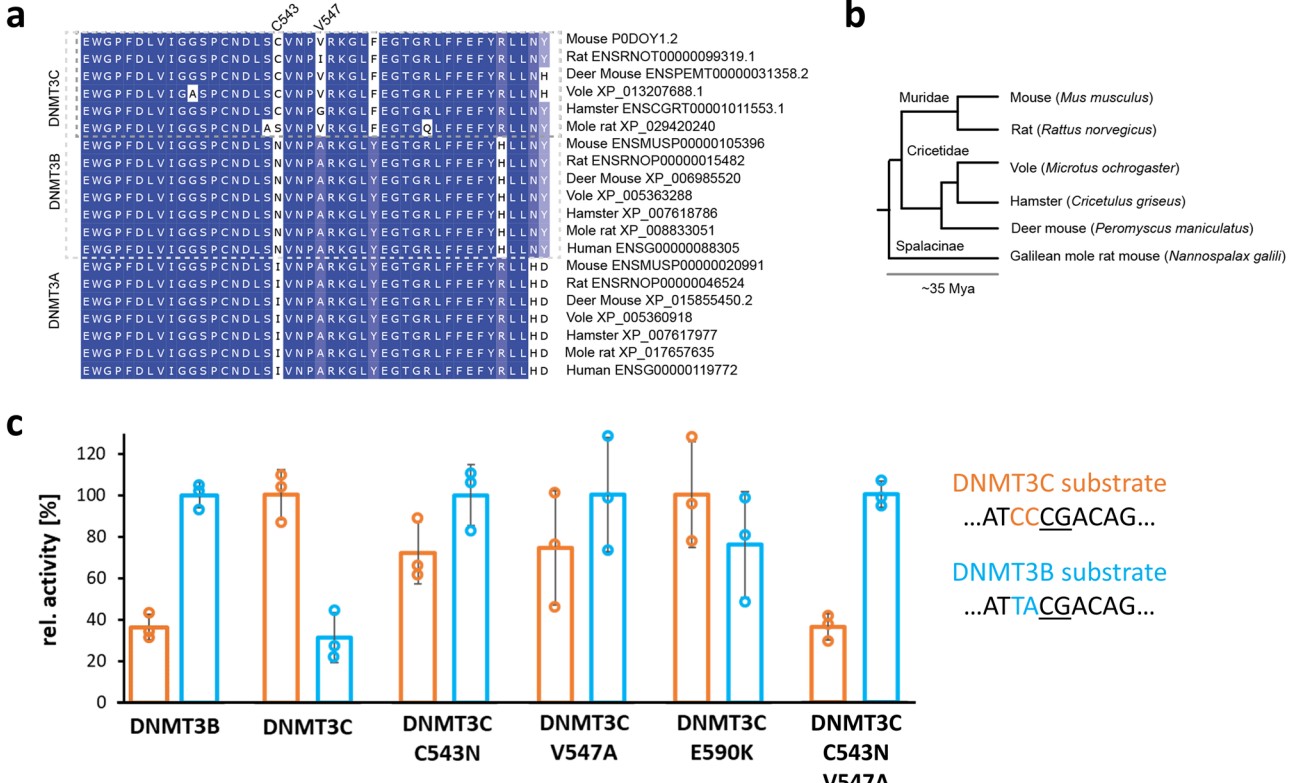

**Fig. 3 | Comparison of the flanking sequence preferences of DNMT3B, DNMT3C and DNMT3C mutants determined with model substrates using radioactively labelled AdoMet. a** Multiple sequence alignment of DNMT3A/B/C orthologues in muroidea and humans, highlighting the region around mouse DNMT3C positions C543 and V547. Amino acid color depicts conservation. **b** Species tree of the selected muroidea species found to contain bona fide DNMT3C orthologues, with topology and divergence time adapted from Steppan and Schenk (2017)[61]. **c** Methylation rates of two designed substrates to be preferred by DNMT3B and DNMT3C. Methylation rates were determined for WT DNMT3B, WT DNMT3C, and DNMT3C mutants C543N, V547A, E590K, and the C543N/V547A double mutant. Shown are averages of three experiments, error bars represent the SD, data points are indicated by circles. The -4 to +4 part of the sequence of the substrates is indicated.

DNMT3 enzyme family, and the C(-1) preference is different from DNMT3B.

### Validation of the DNMT3C flanking sequence preferences

To further validate the C(-2/-1) preference of DNMT3C experimentally, we designed two single CpG site methylation substrates, one with C at position -2 and -1 (CCCG) to be preferred by DNMT3C and one with the same sequence but T at -2 and A at -1 (TACG) to be preferred by DNMT3B. Due to the relatively high methylation activity of DNMT3 enzymes at non-CpG sites, for the specific analysis of the methylation of the central CpG site in the upper DNA strand of these substrates, they have to be used in a hemi-methylated (hm) and fully methylated (fm) version at the central CpG site[27]. The substrates of one hm/fm pair only differ in the methylation state in the upper DNA strand of the central CpG sites, which is not already methylated in the corresponding hm substrates (hence available for enzymatic methylation), but it is already methylated in the fm substrate (hence not available for enzymatic methylation). Therefore, the hm-fm difference in the methylation rates of these substrates specifically reflects the methylation status of the central CpG site in the upper DNA strand, which is embedded in the designed flanking context. These substrates were methylated with radioactively labelled AdoMet and the methylation rates were determined by linear regression of the initial part of the reaction progress curves (Supplementary Fig. 4a). This analysis clearly revealed a preferential methylation of the DNMT3C-substrate by DNMT3C, while the DNMT3B-substrate was better methylated by DNMT3B (Fig. 3c), which is in agreement with the NGS methylation results. Of note, the TACG DNMT3B-substrate was not designed to be disfavored by DNMT3C and it actually contains the residues second most preferred by DNMT3C at the -2 and -1 position.

### DNMT3C-specific flanking preferences are connected to C543 and V547, and evolutionary conserved

To identify the molecular reason for the altered flanking sequence preference of DNMT3C when compared with DNMT3B, we inspected the sequence alignment of DNMT3 enzymes (Supplementary Fig. 1) and the DNMT3B structure (Fig. 1b and d) and searched for amino acid differences in the catalytic domains of DNMT3C and DNMT3B next to the -2 and -1 flanking base pairs of the bound DNA. This analysis led us to the identification of C543, V547, and E590 in DNMT3C as the most plausible candidate residues to mediate the DNMT3C-specific DNA interaction in this region. We prepared, overexpressed and purified the corresponding DNMT3C C543N, V547A, and E590K mutants, in which these residues were exchanged by their counterparts in DNMT3B (Fig. 1c). Next, the activities of the mutants were tested on the designed DNMT3C and DNMT3B methylation substrates (Fig. 3c), showing a reduced preference of E590K for the DNMT3C substrate, but a complete inversion of the substrate preferences for C543N and V547A, which now prefer the DNMT3B substrate over the DNMT3C substrate. These results suggested that C543N and V547A play key roles in the DNA interaction at the -2 and -1 flank region. Therefore, we generated, expressed, and purified a C543N/V547A double mutant (Fig. 1c) and tested it on the designed substrates. Strikingly, the C543N/V547A mutant showed clear DNMT3B-like preferences on these substrates (Fig. 3c). Therefore, the flanking sequence preferences of the DNMT3C C543N/V547A mutant were also tested on the randomized substrate in the Deep Enzymology approach. Following the same workflow as described for wildtype (WT) DNMT3C (Supplementary Fig. 2 and 3d), our data revealed a perfect DNMT3B flanking sequence preference of the DNMT3C C543N/V547A mutant (Fig. 2e and f). This result indicates that

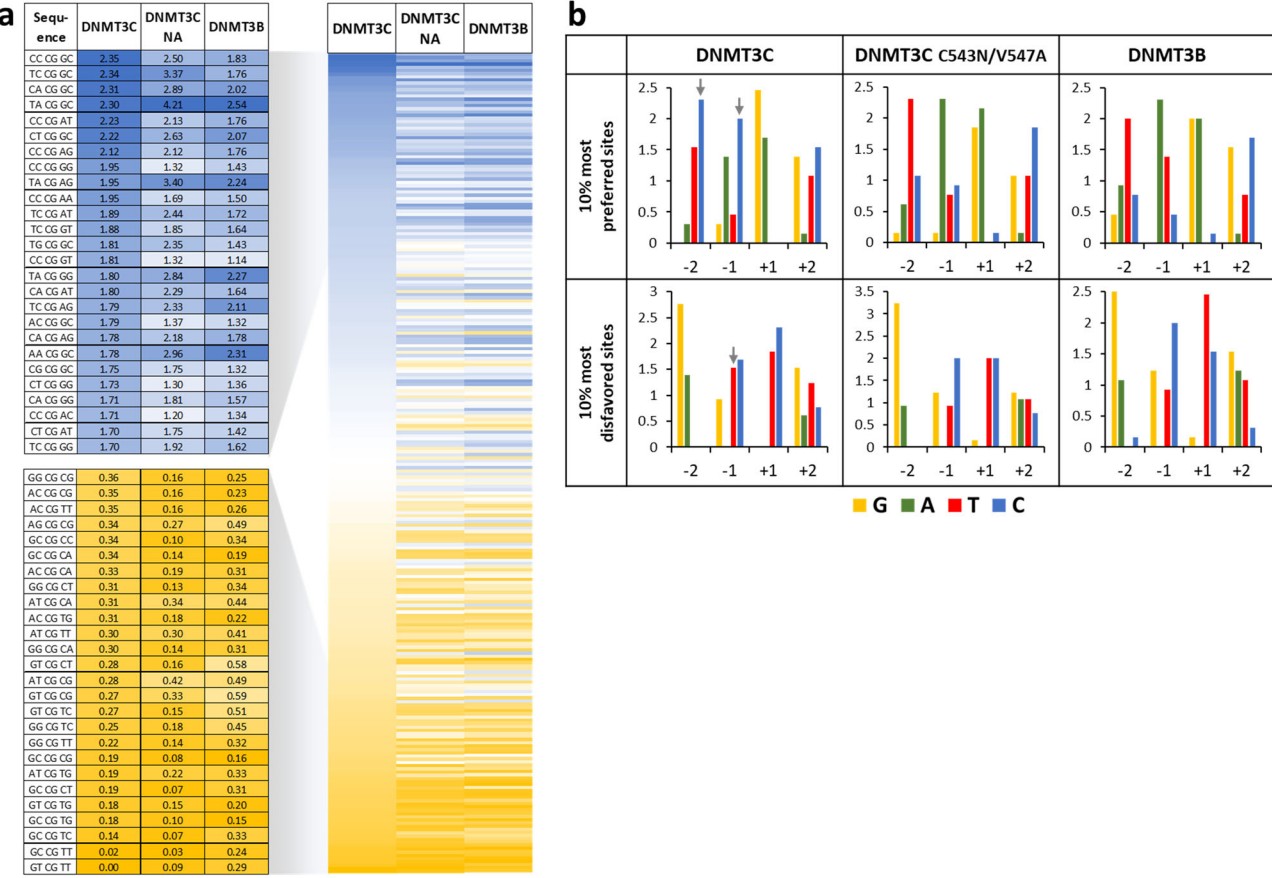

**Fig. 4 | Detailed analysis of NNCGNN flanking sequence preferences of DNMT3C. a** Comparison of the NNCGNN methylation rates of WT DNMT3C, DNMT3C C543N/V547A (NA) double mutant and WT DNMT3B. Examples of the rate fittings are shown in Supplementary Fig. 5. Shown are the 10% most preferred/ disfavored sites. DNMT3B data were taken from[19]. **b** Enrichment and depletion of bases at the -2 to +2 flank of the 10% most preferred and most disfavored NNCGNN sequences. DNMT3C-specific effects at positions -2 and -1 are highlighted by arrows.

the C543 and V547 residues in DNMT3C are responsible for its unique and characteristic C(-2/-1) preference.

A phylogenetic analysis revealed that C543 is fully conserved in the different DNMT3C proteins (Fig. 3a and b). Moreover, Val is the predominant residue at position 547. In rats, V547 is changed to Ile which has similar physicochemical properties and is expected to cause similar biochemical effects as Val. In the hamster, a Gly residue is observed at position 547, which also has the potential to cause structural changes of the entire loop due to its more relaxed backbone conformations. To investigate the consequences of these amino acid changes, we have introduced the V547I and V547G mutations into murine DNMT3C, thereby creating murine DNMT3C variants that mimic the DNMT3C versions in rats and hamsters. Flanking sequence preference analyses were performed in two replicates which correlated very well (Supplementary Fig. 2). The merged data clearly demonstrated that both enzyme variants still exhibit the preference for C at the -2 and -1 positions that is characteristic for DNMT3C (Supplementary Fig. 7).

**Quantitative analysis reveals 100-fold flanking sequence preferences of DNMT3C and higher non-CpG activity**

For a more quantitative analysis of the flanking sequence preferences and potential higher-order dependencies, we extracted average NNCGNN methylation levels in all individual data sets. The methylation of all NNCGNN sites were fitted to 256 monoexponential reaction progress curves using a virtual time axis to extract the corresponding methylation rate constants for DNMT3C and the DNMT3C C543N/V547A mutant (Supplementary Fig. 5). Available murine DNMT3B data[19] were treated in the

same way. Overall, the most (CCCGGC) and least (GTCGTT) preferred NNCGNN flanks of DNMT3C differed more than 100-fold in their methylation rates. Comparison of the methylation rates with the corresponding data from DNMT3B confirmed the previous observations that DNMT3C has a unique flanking sequence preference (Fig. 4a). The magnitude of this effect is illustrated by the finding that the ratio of relative methylation rates of given NNCGNN sites by DNMT3C and DNMT3B differed by almost 30-fold. Highlighting of the most and least preferred NNCGNN sites revealed the common preference of DNMT3C and DNMT3B for T(-2) and A(-1), but a striking increase in preference of DNMT3C for C at -2 and -1, in addition to an increased disfavor for T(-1) (Fig. 4b). In contrast, the DNMT3C C543N/V547A mutant basically reproduced DNMT3B data (Fig. 4).

We also used the CpN methylation data and extracted average methylation levels at CpG, CpA, CpT and CpC sites (Supplementary Table 2). To determine the stringency of CpG recognition, these data were fitted to exponential reaction progress curves and compared with DNMT3B data[28] analyzed accordingly (Supplementary Fig. 6a). Comparison of the methylation rates indicated that the relative CpA and CpT methylation by DNMT3C is about 1.5-fold increased when compared with DNMT3B (Supplementary Fig. 6b), which represents a moderate effect much less than what was observed with some DNMT3A mutants[29]. More detailed analyses revealed that for CpA and CpT methylation the C(-1) preference of DNMT3C is lost, indicating that this DNMT3C-specific effect only manifests in a CpG context (Supplementary Fig. 6c). In addition, at the +1 site the G > A preference is more pronounced in CpA and CpT than in CpG methylation, similarly as previously observed for DNMT3B[28]. In the case of

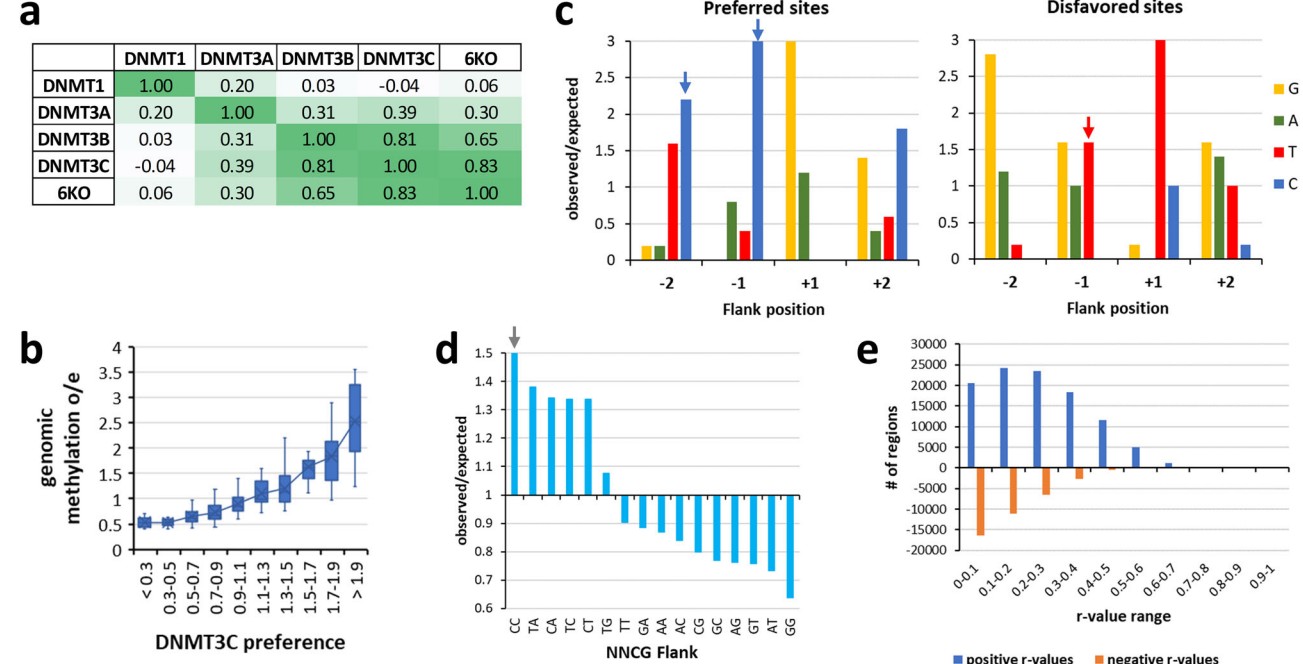

**Fig. 5 | Correlation of DNMT3C flanking sequence preferences and genomic methylation. a** Correlation of the NNCGNN flanking profiles of different DNMTs with the genomic methylation of 6KO cells[33]. DNMT data were taken from: DNMT1[53], DNMT3A[19,29]; DNMT3B[19]. Shown are Pearson *r* values. See also Supplementary Fig. 8 and Supplementary Table 4. **b** Correlation of DNMT3C flanking sequence preferences and genomic methylation in o/e ratios for all 256 NNCGNN sequences. The boxes show the median, 1st and 3rd quartile. Whiskers display the 1.5 IQR distance. X indicates the average. **c** Enrichment and depletion of bases at the -2 to +2 flank of the 10% most highly/lowly methylated NNCGNN sequences in 6KO cells. DNMT3C-specific effects at position -2 and -1 are highlighted by arrows. **d** Methylation levels of NNCG sites in 6KO cells expressed as o/e. The DNMT3C-specific CC preference is highlighted by an arrow. **e** Local correlation analysis of methylation levels and DNMT3C NNCGNN preferences of consecutive CpG sites in chromosome 1 of 6KO. R-values were determined in a 22 CpG site window, positive r-values are shown in blue, negative ones in orange. Examples of local correlations are shown in Supplementary Fig. 9.

CpT, no methylation was seen at ATCTAA and ACCTTT sites, despite good sequence coverage of these sites in the NGS data suggesting that some structural features of these sequences prohibit DNMT3C activity.

## The DNMT3C flanking sequence preference determines its activity in mouse ES cells

In order to investigate if the newly determined DNMT3C CpG methylation flanking sequence preferences affect cellular CpG methylation patterns, we reanalyzed recently published genome-wide DNA methylation data from 6KO mouse ES cell lines containing deletions of all other DNMT (DNMT1, DNMT3A, DNMT3B) and TET (TET1, TET2, TET3) enzymes[33]. In these cells, DNMT3C is the only DNA methylation-related enzyme, hence we expected clear-cut results, which are not confounded by effects of other enzymes. Average NNCGNN methylation levels were extracted, revealing a mean methylation level of 1.46% in the 6KO cells, which was almost completely lost after additional removal of DNMT3C (7KO cells) (Supplementary Fig. 8). The NNCGNN methylation profile of the 6KO cells was compared with the corresponding profiles of DNMTs, revealing a very strong (r = 0.83) correlation with DNMT3C (Fig. 5a and b). Analysis of the sequences of the most preferred and disfavored NNCGNN sites revealed a pattern that almost mirrors the DNMT3C profile including preferences for C(-2) and C(-1) and disfavor of T(-1) (Fig. 5c). The very strong correlation between genomic methylation in the 6KO cells and the biochemical flanking sequence preferences of DNMT3C is also indicated in the o/e frequencies of genomic methylation in NNCG sites. In this context, CC was the most preferred dinucleotide at the NN positions followed by TA, CA, TC, and CT, which corresponds to different perturbations of C and T at -2 and C and A at -1 that are most preferred by DNMT3C (Fig. 5d).

To further validate the correlation of CpG methylation levels in 6KO cells and DNMT3C NNCGNN preferences, local correlation analysis of consecutive CpG sites were conducted with a sliding window of 22 sites in chromosome 1. As shown in Fig. 5e, positive correlations were observed much more frequently than negative correlations. The probability of this distribution was tested by randomizing the NNCGNN preferences and repeating the analysis, revealing a Z-value of >11 corresponding to a *p* value < 10^-99. Examples of local correlations are shown in Supplementary Fig. 9, illustrating that in each case, genomic methylation peaks were observed at peaks of the relative NNCGNN preferences of DNMT3C.

## DNMT3C flanking sequence preference facilitates methylation of retrotransposon known to be DNMT3C targets

Our data presented so far demonstrate that DNMT3C shows a specific increase in flanking sequence preference for CpG sites with C at the -2 and -1 flank sites, whereas these sites are not preferred by DNMT3B. As DNMT1 is present in cells and ensures efficient methylation of hemimethylated CpG sites, it is not important if an initial methylation is introduced into the upper or lower DNA strand. Hence the biologically relevant DNMT3C flanking sequence preference corresponds to CCCG or CGGG. Loss of DNMT3C was shown to cause an increased expression of non-LTR (mainly L1Md_A, L1Md_T and L1Md_Gf) and ERVK elements (for example, MMERVK10C, IAPEz, and IAP_d)[12]. We retrieved the sequences of corresponding L1Md and IAP elements from the Dfam database[34] and analyzed them for the occurrence of CpG sites in a CCCG and CGGG context. As expected for young retrotransposons, all sequences are rich in CpG sites with o/e values > 1 in many cases (Supplementary Table 5). Even the lowest CpG o/e ratio of 0.45 is much above the corresponding average of the mouse genome of 0.192[35]. As illustrated in Fig. 6a for L1Md_A and IAPEz and summarized in Supplementary Table 5, we noticed a strong enrichment of CpG sites in the promoter regions of all three L1 elements (the CpG o/e value for the entire mouse genome is 0.192[35]). In these regions, the frequency of CpG sites in the CCCG or CGGG context was also massively increased (using the nucleotide composition of the corresponding repeat element as a reference).

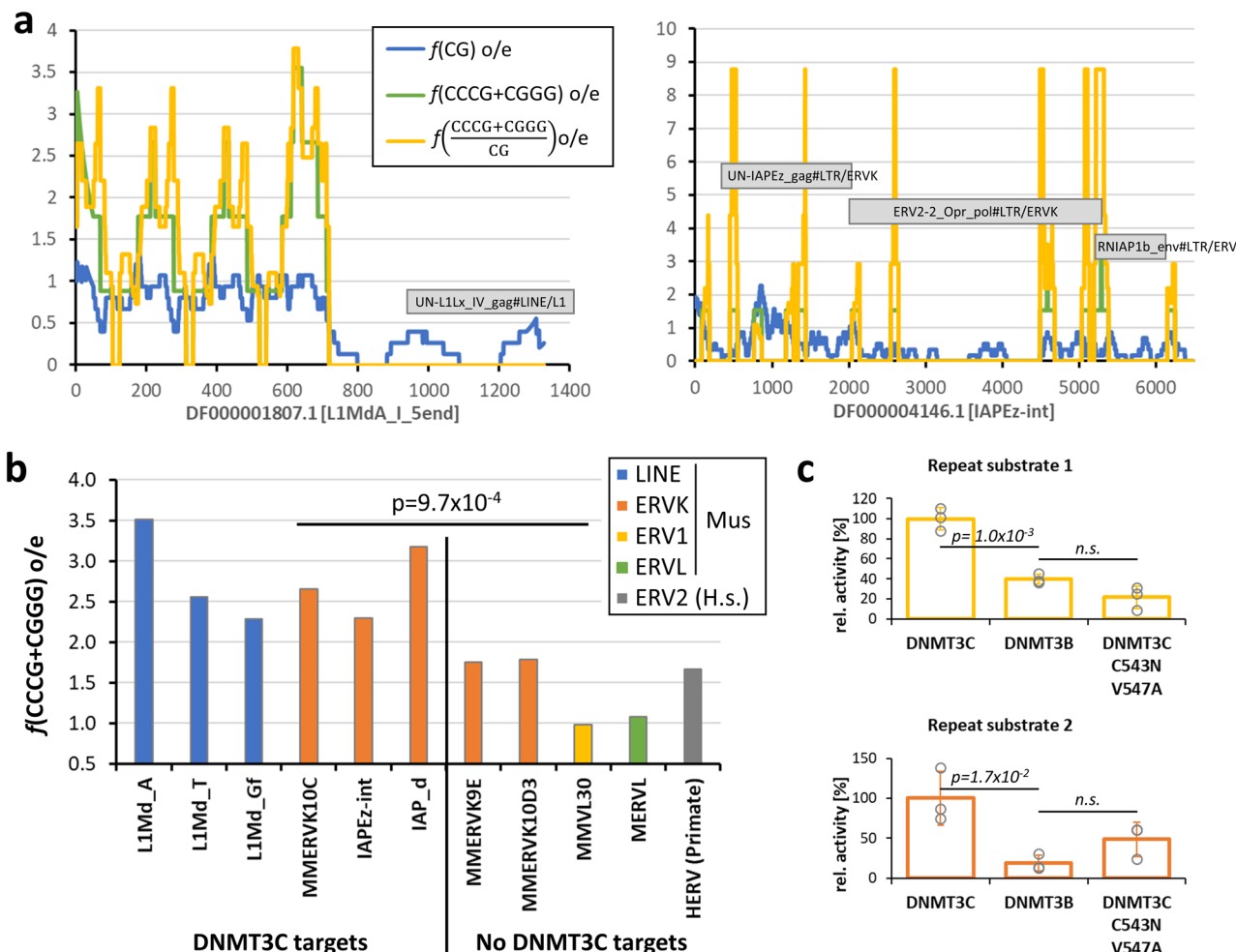

**Fig. 6 | Enrichment of CCCG or CGGG sequences in mouse repeat elements that are DNMT3C targets. a** Image of the L1MdA and IAPEz-int elements showing the frequencies of CpG sites, CCCG + CGGG sites and fraction of CpG sites in CCCG or CGGG context, all expressed as o/e frequencies averaged over 50 bps. "Expected" values were determined using the C and G content of the entry. The annotated ORFs of the elements are indicated. Note the clustering of CCCG/CGGG sites in the promoter of L1MdA and at several regions in the body of IAPEz. **b** Frequency of CG sites in a CCCG or CGGG context in mouse repeat elements. The figure shows data for three LINE elements and three ERVK elements that are validated DNMT3C targets[12]. As a control, two ERVK elements are shown, which are not methylated by

DNMT3C. In addition, data are shown for examples of one murine ERV1 and ERVL element that both are also not targeted by DNMT3C and for one human ERV2 element. For the L1 elements, the promoter regions were used for the analysis, in all other cases the entire sequence of the element was used. See Supplementary Table 5 and Supplementary Fig. 10 for more information. The *p*-value was determined by a two-flanked T-test assuming unequal variance. **c** Methylation rates of two designed repeat substrates with sequences taken from known DNMT3C targets. Shown are averages of three experiments, error bars represent the SD. Data points are indicated by circles. P-values were determined by a two-sided T-test assuming equal variance, n.s. not significant. See also Supplementary Fig. 4.

Peak o/e values of CCCG/CGGG are in the range of 2-3.7 when averaging the frequencies over 50 bps (Fig. 6a) and the overall o/e over the entire promoter region was 3.5 (Fig. 6b, Supplementary Fig. 11 and Supplementary Table 5). Similar effects with o/e values of 2.6 and 2.3 were observed in the promoters of L1Md_t and M1Md_Gf elements. As illustrated for IAPEz (Fig. 6a and b) and IAPd (Fig. 6b), strong enrichment of CCCG/CGGG sites is also observed in the entire IAP elements that are targeted by DNMT3C. In these elements, CCCG/CGGG sites are clustered at several regions (Fig. 6b and Supplementary Fig. 12), future work will be needed to determine the functional roles of these sites. Strikingly, the CCCG/CGGG sequence is found in the consensus binding motifs of important transcription factors, e.g. of E2F6, GCM1 and SP2, indicating its potential biological relevance (Supplementary Fig. 11). In summary, this analysis indicates that CCCG/CGGG sites are highly enriched in retrotransposon that have been identified to be targets of DNMT3C[12].

Next, we aimed to determine the CCCG/CGGG content in negative control repeat elements. For this, we inspected the published data[12] for elements that did not show DNMT3C dependent changes in DNA

methylation and expression and for which DFAM entries were available with sufficient length (>400 bps) and identified two ERVK elements that were suitable for our analysis (MMERVK9E and MMERVK10D3). Both elements showed similar enrichment in CpG sites as the ERVK elements that are DNMT3C targets (Supplementary Table 5). However, the level of CpG sites in CCCG/CGGG context was significantly smaller than in the DNMT3C targets (*p*-value $9.7\times10^{-4}$ based on two sided Ttest with unequal variance) (Fig. 6b and Supplementary Table 5). Murine ERV elements from other classes (ERV1, ERVL) that are not targeted by DNMT3C, and a human ERV2 element showed low CCCG/CGGG frequencies as well. We conclude that validated DNMT3C targets exhibit specific enrichment of CpG sites in CCCG/CGGG context either in their control regions or throughout the entire element.

These observations suggest that the DNMT3C flanking sequence preferences facilitate the methylation of retrotransposons that were shown to be biological targets of DNMT3C. To validate this conclusion, we designed two substrates on the basis of natural sequences taken from these sources. As described above, the substrates needed to be used as hm/fm

pairs, hence only single CpG site substrates were suitable. We managed to design one 30-base pair substrate entirely on the basis of the L1Md_a sequence (repeat substrate 1). For a second repeat substrate, 12 base pairs were taken from L1Md_t (repeat substrate 2) and embedded into the sequence scaffold used by us before. For normalization of the absolute activities of DNMT3C and DNMT3B, a long DNA substrate containing 38 CpG sites was used (414-mer) that was equally preferred by both enzymes (average relative site preference: DNMT3C = 1.33, DNMT3B = 1.39). The methylation data revealed a clear and highly significant increase in the relative activity of DNMT3C on the repeat substrates that was lost in the DNMT3C C543N/V547A mutant (Fig. 6c, Supplementary Fig. 4b). In summary, these data indicate that the flanking sequence preferences of DNMT3C are supporting the methylation of young retrotransposons, the known biological target of DNMT3C in the murine male germline.

## Discussion

Active retrotransposons are known to impair fertility in mammals, and silencing of these elements is necessary during the early embryonic stages and in germline development[36]. Hence, silencing of repeat elements is an essential process and several dedicated (epi)genetic systems are involved in it. This includes the KRAB-ZNF silencing system, which targets repeats with specific zinc finger proteins[37,38] and delivers H3K9me3 via SETDB1[39], DNA methylation via DNMT3A[40] and HDAC activity[41]. The piRNA system also delivers H3K9me3 and DNA methylation to repetitive elements targeted by repeats-derived piRNAs[42,43]. Previous work has shown that DNA methylation by DNMT3A and DNMT3L plays very important roles in the piRNA-mediated methylation of repeat elements in murine male germline cells[15,30,44,45]. Additionally, in muroids, a duplication of the *DNMT3B* gene has given rise to a novel DNMT3 ortholog, called DNMT3C, that is expressed in the male germline and was shown to be required for the methylation and repression of young non-LTR retrotransposons and ERVK elements[12,13]. However, the specific properties and adaptations connecting DNMT3C to these classes of repeat elements have remained unclear.

As previous work showed that flanking sequence preferences of DNMT3 enzymes co-evolved with their biological targets[19,28], we investigated the hypothesis that DNMT3C might display flanking sequence preferences that match to its specific biological targets as well. Indeed, we identified a shift in the flanking sequence preference profile of DNMT3C that is distinct from the preferences of DNMT3A and DNMT3B and shows a unique preference for C at the -2 and -1 flank sites. Analysis of the sequence alignment of DNMT3 enzymes and structures of DNMT3B[19] led to the identification of two residues (C543 and V547) in the catalytic loop of DNMT3C that mediate the DNMT3C-specific DNA interaction at the -2 and -1 flanking base pairs. This loop immediately follows the key catalytic PCN motif[46], which explains its high relevance for controlling catalytic activity. The exchange of the C543 and V547 residues to their DNMT3B counterparts completely reverted the flanking sequence preferences of the DNMT3C C543N/V547A double mutant back to the original DNMT3B pattern. A phylogenetic analysis showed that C543 is fully conserved in muroid DNMT3C enzymes, and at position 547 Val is predominant, but Ile and Gly are also observed in the DNMT3C orthologs of rats and hamsters. Additional experimental data demonstrated that DNMT3C enzymes carrying Ile or Gly at position 547 show the same flanking sequence preferences as murine DNMT3C, indicating that these effects are conserved in evolution.

These findings can be explained from a structural point of view. N656 in DNMT3B (the residue corresponding to DNMT3C C543) is placed in the major DNA groove at the -2/-1 flank region (Fig. 1d). Its exchange by Cys in DNMT3C (with much reduced H-bonding potential) is expected to alter the hydration of the major groove and disrupt DNMT3B-specific DNA interactions. A660 in DNMT3B (the residue corresponding to DNMT3C V547) points towards the major DNA groove at the +1/+2 flank region, and by this it anchors the entire catalytic loop (Fig. 1d). Exchange of this residue by the larger Val is expected to lead to changes in the loop position and geometry that may also disrupt DNMT3B-specific DNA interactions. Our

finding that the DNMT3C-specific C(-2/-1) preference is only observed when methylation occurs in a CpG context can be rationalized from a structural point of view as well, because two other residues in this loop (DNMT3B V657 and DNMT3B N658) have been shown to modulate the CpG-dependent flanking sequence preferences in DNMT3B[28].

As described so far, we have shown that DNMT3C enzymes exhibit a specific C(-2/-1) preference that is conserved in evolution and caused by two specific amino acid exchanges. In the next step of our analysis, we showed that CCCG/CGGG sites, which are matching the DNMT3C flanking sequence preferences, are enriched in repeat elements known to be targeted by DNMT3C. One potential caveat of our study is that the flanking sequence analysis was conducted using only the catalytic domain of DNMT3C. However, the relevance of the biochemical DNMT3C flanking sequence preferences has been clearly documented by their impressive correlation with cellular DNA methylation patterns observed in ES cells containing endogenous DNMT3C as active DNMT. In addition, future work may aim to investigate the biological role of the methylation of the clustered CCCG/CGGG sites in the repeat elements. As mentioned above, the enrichment of this motif in TF binding sites may suggest that the binding of these factors could be modulated by DNA methylation.

Other factors are also known to affect the regulation and function of DNMTs in cells. In addition, to the mutations in the catalytic domain, DNMT3C has lost the PWWP domain, which normally recruits DNMT3 enzymes to chromatin carrying H3K36me2/3[27,47,48]. In the case of DNMT3A, the PWWP domain has been shown to control chromatin interaction, because inactivating the PWWP domain leads to massive genome-wide retargeting[49–51]. In addition, other DNMTs affect repeat element methylation, in particular DNMT1, which is needed to maintain and propagate the DNA methylation. Our previous work has shown that DNMT1 also exhibits flanking sequence preferences[32,52]. However, in cells containing DNMT1, modulation of methylation levels by DNMT1 flanking sequence preferences occurs at high methylation levels, indicating that most cells have enough DNMT1 activity to methylate even disfavored targets. Nevertheless, insufficient maintenance methylation tends to occur at sites with a flanking sequence context that is disfavored by DNMT1. Interestingly, DNMT1 has a disfavor for C(-2)[52,53], which may have increased the evolutionary pressure for the generation of a DNMT3C-type enzyme.

Based on our data, we propose a co-evolution model of DNMT3C and repeat elements in which repeat elements triggered the evolutionary fixation of the *DNMT3C* gene and afterwards DNMT3C with its new flanking sequence preferences shaped the sequences of repeat elements. According to the presence of DNMT3C in extant muroidea, DNMT3C originated as a duplication of DNMT3B in the muroid ancestors, sometime from 35 to 45 Mya[16]. In Eumuridae, the DNMT3C enzyme acquired the characteristic C543/V547 amino acid signature leading to the specific and novel flanking sequence preferences discovered in our work with a preference for C at the -2 and -1 flank site. Based on the extant biological role of DNMT3C[12,13], it is plausible to postulate that the fixation of DNMT3C and appearance of the C(-2/-1) preference was triggered by evolutionary pressure coming from transposable elements with many CCCG/CGGG sequences in their control elements that were poorly methylated by DNMT3A and DNMT3B. After the establishment of DNMT3C with its novel flanking sequence preference and its expression in germline cells, it could mediate a better protection against transposable elements with CCCG/CGGG sequences. Hence, outbreaks of such elements had a higher chance to be confined not causing the death of the affected germ cell or embryo. Therefore, these elements were better tolerated, and as a consequence, these sequences are enriched in murine young repeat elements, which nowadays are kept neutral with the help of DNMT3C in mice. By this, our data provide a unique mechanistic insight into the molecular co-evolution of repeat elements and (epi)genetic defense systems dedicated to maintain genomic stability in mammals. This process has conceptual similarity to the well-documented co-evolution of KRAB–zinc finger proteins and endogenous retroelements in mammals[54]. These systems recognize repeat elements by sequence-specific zinc finger proteins and via KAP1 they recruit silencing cofactors, which deliver

H3K9me3 and DNA methylation[55]. However, the methylation analyses in DNMT3C KO germ cells suggest that DNMT3C[12,15] has no role in this defense system.

## Methods

### Cloning and protein purification

An *E. coli* codon-optimized gene of the catalytic, C-terminal domain of murine DNMT3C (amino acid residues 439-740 of Uniprot entry P0DOY1) was obtained from IDT (Integrated DNA Technologies). The gene fragment was cloned into the TOPO-TA vector pSC-A-amp/kan from the StrataClone PCR Cloning Kit (Agilent Technologies). For plasmid DNA isolation, the NucleoSpin® Plasmid Kit (MACHEREY-NAGEL) was used. Next, the DNMT3C fragment was cloned into the pET28a(+) expression vector (Novagen) as an N-terminal His$_6$-tag fusion using the Gibson Assembly method. The mutations C543N, V547A, E590K and C543N/V547A (NA) were introduced into DNMT3C by using site-directed mutagenesis[56]. All DNA sequences were confirmed by Sanger DNA sequencing (Microsynth Seqlab GmbH). The WT DNMT3C-C-terminal domain and its mutated variants were over-expressed in BL21 (DE3) codon plus RIL cells (Agilent Technologies). The cells were grown in LB medium until an OD$_{600}$ of 0.6 was reached. The protein expression was induced by the addition of 0.5 mM iso-propyl-β-D-1-thiogalactopyranoside. Afterwards, the cultures were incubated for 15 h at 20 °C with horizontal shaking. The proteins were purified using nickel-nitrilotriacetic acid-agarose and stored in 20 mM HEPES, pH 7.5 supplemented with 200 mM KCl, 0.2 mM DTT, 1 mM EDTA, and 10% glycerol at -80 °C. Some precipitation of the protein occurred during dialysis for the WT DNMT3C as well as for mutant proteins. The protein purity and concentrations were determined using Coomassie-stained 12% SDS polyacrylamide gels. The catalytic domain of murine DNMT3B (amino acid residues 558–859 of Uniprot entry O88509) was purified as described for the human protein[28].

### Radioactive DNA methylation activity assay

Enzymatic activities of DNMT3C and DNMT3B catalytic domain and the DNMT3C mutants were analyzed by avidin-biotin methylation plate assay using biotinylated double-stranded 30-mer oligonucleotides with a single CpG site basically as described[57]. Two substrates were used, one optimized for DNMT3B (TACGAC sequence context) and one for DNMT3C (CCCGAC sequence context) as specified in Supplementary Table 3 together with the other substrate sequences that were used (repeat 1, repeat 2 and a 414-mer substrate/oligonucleotide). As DNMT3 enzymes also show activity at cytosine residues in non-CpG context, the specific methylation of the central CpG in the upper DNA strand was determined by comparison of methylation activity of fully methylated double-stranded substrates and hemimethylated substrates carrying a pre-methylation in the lower DNA strand as described in the results section. The methylation reactions were performed at 37 °C in 1x methylation buffer (20 mM HEPES pH 7.5, 1 mM EDTA, 50 mM KCl, 0.25 mg/mL bovine serum albumin) using 1 µM of biotin-tagged substrate. Methylation reactions were conducted using 9.8 µM WT DNMT3C, 7.6 µM C543N mutant, 10.8 µM V547A mutant, 14.0 µM E590K mutant, 10.4 µM C543N/V547A mutant and 10.4 µM WT DNMT3B and started by adding 0.76 µM radioactively labeled [methyl-³H]-AdoMet (Perkin Elmer). Samples were collected in duplicates at time points of 2, 4, 8, and 12 min. Methylation rates were determined by linear regression of the incorporated radio-activity during the inital reaction phase. All reactions were conducted at least in triplicates.

### Analysis of flanking sequence preference with a randomized substrate

For analysis of the flanking sequence preference, substrates with CpG or CpX sites in a 10 base pairs randomized sequence context were prepared as described[28,29]. Substrate methylation reactions were performed with different enzyme concentrations of WT DNMT3C or C543N/V547A mutant (1.2–20.8 µM) in 1x methylation buffer (20 mM HEPES pH 7.5, 1 mM EDTA, 50 mM KCl, 0.25 mg/mL bovine serum albumin) containing 1 mM AdoMet (Sigma-Aldrich) and 1 µM randomized substrate. Methylation reactions were incubated at 37 °C for 30–60 min. The reactions were stopped by freezing in liquid N$_2$ and then incubated with proteinase K (80 U/µL, NEB) for 1.5 h at 42 °C and 300 rpm. The DNA was purified with PCR Clean-up kits (Macherey-Nagel). Processing, hairpin ligation and bisulfite conversion was conducted in principle as described[53]. Hairpin ligation was performed using 1 U/µL BsaI-HF®v2 (NEB) and 36 U/µL T4 DNA ligase (NEB) with a specific single-stranded hairpin oligonucleotide (27mer) to allow for the reconstitution of the randomized sequence after bisulfite conversion. Subsequently, bisulfite conversion was performed using the standard protocol from EZ DNA Methylation-Lightning™ Kit (ZYMO RESEARCH) with final elution in 10 µL RNase free water[53] to distinguish between methylated and non-methylated cytosine bases in the methylation products. Library preparation for next-generation sequencing (NGS) was performed as described[53] with two PCRs using variable primer pairs to introduce sample-specific barcodes and indices for sample distinction and the sequencing reactions[6]. Illumina NGS was conducted by Novogene Company Limited (Cambridge, UK).

### Bioinformatic data analysis of the methylation of randomized substrates

Bioinformatic analysis of the NGS data was carried out basically as described[28,29] using a local Galaxy server[58]. For this, sequenced fastqsanger files were uploaded, trimmed with the Trim tool (Galaxy Version 0.0.1) to a desired length of 128 nucleotides, and filtered to a quality score of minimum 20 using Filter FASTQ tool (Galaxy Version 1.1.1)[59]. Next, duplicates were removed using a local tool (developed by Pavel Bashtrykov). For this, sequences were sorted alphabetically and identical sequences were removed in order to avoid artificially changes in the statistics by increasing the occurrence of individual sequences. Considering the sequence information of the upper and lower DNA strands, the methylation state of the CpG/CpX site and the original sequence of the randomized parts were determined using a home-written script in Galaxy (developed by Pavel Bashtrykov). First the methylation of the central cytosine is analyzed, which allows a division into methylated and unmethylated reads. Then, the frequency of all bases at each -8 to +8 positions is calculated for the methylated and unmethylated sequences and enrichment/depletion ratios in the pool of methylated sequences are calculated for each base at each position. For substrates with a CpX motif, methylation within CCWGG sequences was excluded from the analysis, because it is overlapping with the *E. coli dc*m methyltransferase that is expressed in the protein overexpression BL21 (DE3) codon plus RIL strain[28,29]. Pearson r-values of the correlation of -8 to +8 flanking sequence preference profiles were created using Microsoft Excel. In addition, average methylation levels of all NNCGNN flanking sequences were determined and furhter analysed using Microsoft Excel. For determination of the methylation rates of all 256 NNCGNN sequences by one enzyme, methylation reactions of individual substrates were assumed to be independent and reaction velocities are of first order with respect to the substrate concentrations. The results of the individual reactions with different enzyme concentrations and incubation times were fitted to monoexponential reaction progress curves using variable virtual time values. Fitting was conducted with MatLab as described, except that convergence was validated by serial fitting[29].

### Genomic methylation data analysis

DNA methylation in wildtype murine ES cells as well as multiple DNMT and TET KO cell lines was investigated using whole genome bisulfite data published by Wang et al. 2020 (GEO accession number GSM3239884 and GSM4809269)[33]. Supplementary files containing methylation calls uploaded by authors were used for the analysis. Genomic sequences of 22 nt with central CpG sites were obtained, and CpG sites with a sequencing coverage ≥5 were used to determine average methylation levels in all NNCGNN flanking contexts.

## Sequence search and alignment

The mouse DNMT3C protein sequence was used to gather orthologues in other rodent genomes by searching ENSEMBL pre-computed orthologues and NCBI non-redundant database using BLASTp. The best hits were aligned together with DNMT3A/B representatives of the same species using MAFFT with L-INS-i parameters[60]. The alignment was visualized using UGENE software. The species phylogenetic tree of rodents was adapted from Steppan and Schenk (2017)[61].

## Statistics and reproducibility

Pearson correlation factors (r-values) were determined with Excel correl. P-values for the enrichment of sequence elements were determined by binomial statistics using Excel binom.dist. The significance of correlations was tested by randomization of one data set at least 20 times, followed by determination of the r-values, their average, and standard deviation. Based on this, the significance of the original r-value was determined by Z-statistics using Excel norm.s.dist. The $p$ values for averages of methylation rates were determined by two-sided T-test assuming equal variance with Excel.

## Data availability

NGS kinetic raw data are available at DaRUS at https://doi.org/10.18419/darus-3386. Uncropped images and source data of all figures are available in Supplementary Data 1.

## Code availability

The code used for genomic DNA methylation analysis is available at: https://doi.org/10.6084/m9.figshare.23675400.v1[62].

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

## Acknowledgements

This work has been supported by the Deutsche Forschungsgemeinschaft (JE 252/10 and JE252/36). AdM is supported by a European Research Council Starting Grant number 950230.

## Author contributions

L.D. conducted all experiments included in this study. P.B. contributed to the bioinformatic work and did the genome methylation data analyses. A.dM. provided the phylogenetic analysis. L.D., M.E., P.B., A.dM. and A.J. were involved in data analysis and interpretation. M.E. and A.J. supervised the study. L.D. and A.J. prepared the figures and first draft of the manuscript. M.D. provided reagents. All authors approved the final version of the manuscript.

## Funding

## Competing interests

The authors declare no competing interests.
