## [peer review file · Communications Biology]

Reviewers' comments:

Reviewer #1 (Remarks to the Author):

Recent publications, including several from the authors here, have demonstrated that DNMT3 enzymes have a sequence preference for nucleotides flanking the CpG that influence the catalytic activity and subsequent methylation patterns. This manuscript describes an analysis into the flanking sequence preferences for DNMT3C, the murine specific DNA methyltransferase that targets young retrotransposons for silencing.

The study uses a nice complement of approaches to investigate the flanking sequence preferences for DNMT3C. In general the results are well presented and convincing, with a couple of exceptions that are detailed below.

Overall, the findings described here are important for understanding the epigenetic regulation of repetitive elements in rodent genomes and provide interesting perspectives on the evolution of epigenetic regulation.

Specific points:

1. The analysis presented in Figure 2B suggests, at least to this reviewer, that the first preference for DNMT3C is a C at -2 and an A at -1 (CACG). However, the authors state that the first preference is C at both -2 and -1 (CCCG). Is this simply to distinguish the preference of DNMT3C to that of DNMT3B, which prefers an A at -1?

2. The paragraph on the top of page 6 (lines 177-190) has a number of typos. C543 is referred to as C534 in a number of places, which is easy enough to sort out. However, lines 180-182 have a sentence stating "... (Figure 3C), showing a reduced preference of E590K for the DNMT3C substrate, but a complete inversion of the preferences for C534N and V547A ..." Did the authors mean DNMT3B substrate? That would make more sense to me given the data shown in Figure 3.

The results section could furthermore benefit from more detail being included in this manuscript. The authors note that they used "an established workflow based on hairpin ligation, bisulfite conversion, NGS and bioinformatic analysis" and cite references 18,27,28, but it would be helpful to have more detail contained here.

3. The model of co-evolution of DNMT3C and specific subfamilies of repetitive elements is interesting. It would be helpful for the authors to put this argument in the context of what is known about the co-evolution of epigenetic machinery and repetitive elements (ie, the "arms-race" between KZFPs and repetitive elements). Are there unique features of these subfamilies that are DNMT3C targets compared to other subfamilies not targeted by DNMT3C?

As a minor point, the only reference to Figure 3D is at the very end of the Results well after the other figures/panels have been discussed.

Reviewer #2 (Remarks to the Author):

This is an interesting paper in which the authors applied specialized assay systems to investigate the flanking sequence preferences of DNMT3C and observed characteristic preferences for cytosine at the -2 and -1 flank of DNMT3C. The data showed DNMT3C flanking sequence preference matches the sequences of young murine retrotransposons thus facilitating their methylation. In general, this paper provides a new insight towards how the specificity of an epigenetic enzyme facilitates it in regulating retrotransposons silencing.

Below are some detailed comments:

1. DNMT3C flanking sequence preference has been proved to affect cellular CpG methylation pattern in mouse ES cells. One question is whether in the absence of DNMT3C, these retrotransposons can still be methylated? Or the function can be rescued by other DNMT3s?
2. In Figure 3C and D, C543 and V547 mutants are used to study the methylation rates on two designed substrates with TA and CC respectively. How about other sequence combination for the substrates? This can be tried to obtain a more credible conclusion.
3. Another question is for this specific binding preference, is it due to DNMT3C alone or the interaction with other members of the DNMT family, such as DNMT3A and DNMT3B?
4. It is mentioned that the binding of DNMT3C to retrotransposons enhances their methylation and silencing during early development. so is the expression pattern of DNMT3C having any developmental stage or tissue specificity? It would be beneficial to investigate the differences in expression dynamics between DNMT3C and other DNMT3s in different developmental stages.
5. Are there any DNMT3C ChIP-seq previously been performed? Can authors make use of ChIP-seq data to further confirm whether binding specificity of DNMT3C is also correlated with the specified flanking sequences in the genome?

Overall, I think this is a nice investigation on the sequence preference for DNMT3C, and the authors also extend the finding to further hypothesize how this differed flanking sequence preference affects specific retrotransposon silencing. It provides novel insights for future investigation on the sequence-specific silencing of retrotransposons during development. It also promotes us to look into the structural basis of different DNMT3 enzymes for their sequence preferences. Therefore, I think this paper has demonstrated value for the current research field.

Response to the Reviewers' comments

Reviewer #1

“Recent publications, including several from the authors here, have demonstrated that DNMT3 enzymes have a sequence preference for nucleotides flanking the CpG that influence the catalytic activity and subsequent methylation patterns. This manuscript describes an analysis into the flanking sequence preferences for DNMT3C, the murine specific DNA methyltransferase that targets young retrotransposons for silencing.

The study uses a nice complement of approaches to investigate the flanking sequence preferences for DNMT3C. In general the results are well presented and convincing, with a couple of exceptions that are detailed below.

Overall, the findings described here are important for understanding the epigenetic regulation of repetitive elements in rodent genomes and provide interesting perspectives on the evolution of epigenetic regulation.”

Reply: Thank you very much for this positive assessment.

“Specific points:

1. The analysis presented in Figure 2B suggests, at least to this reviewer, that the first preference for DNMT3C is a C at -2 and an A at -1 (CACG). However, the authors state that the first preference is C at both -2 and -1 (CCCG). Is this simply to distinguish the preference of DNMT3C to that of DNMT3B, which prefers an A at -1?”

Reply: Thank you for pointing out an unclear writing at this part. The corresponding paragraph has been rewritten.

“2. The paragraph on the top of page 6 (lines 177-190) has a number of typos. C543 is referred to as C534 in a number of places, which is easy enough to sort out. However, lines 180-182 have a sentence stating “... (Figure 3C), showing a reduced preference of E590K for the DNMT3C substrate, but a complete inversion of the preferences for C534N and V547A ...” Did the authors mean DNMT3B substrate? That would make more sense to me given the data shown in Figure 3.”

Reply: We apologize for the typo in the numbers, which has been corrected now. The other sentence has been rewritten for clarity.

The results section could furthermore benefit from more detail being included in this manuscript. The authors note that they used “an established workflow based on hairpin ligation, bisulfite conversion, NGS and bioinformatic analysis” and cite references 18,27,28, but it would be helpful to have more detail contained here.

Reply: We have expanded the methods description accordingly.

“3. The model of co-evolution of DNMT3C and specific subfamilies of repetitive elements is interesting. It would be helpful for the authors to put this argument in the context of what is known about the co-evolution of epigenetic machinery and repetitive elements (ie, the “arms-race” between KZFPs and repetitive elements). Are there unique features of these subfamilies that are DNMT3C targets compared to other subfamilies not targeted by DNMT3C?”

Reply: We have added a sentence pointing to the similarities of the DNMT3C divergence and co-evolution of KRAB zinc finger proteins and repeat elements. The methylation data in DNMT3C KO germ cells (Barau et al. 2016, Dura et al. 2020) suggest that DNMT3C has no role in this process. This information has been added to the discussion as well.

“As a minor point, the only reference to Figure 3D is at the very end of the Results well after the other figures/panels have been discussed.”

Reply: Figure 3C has been shifted to 6C.

Reviewer #2

“This is an interesting paper in which the authors applied specialized assay systems to investigate the flanking sequence preferences of DNMT3C and observed characteristic preferences for cytosine at the -2 and -1 flank of DNMT3C. The data showed DNMT3C flanking sequence preference matches the sequences of young murine retrotransposons thus facilitating their methylation. In general, this paper provides a new insight towards how the specificity of an epigenetic enzyme facilitates it in regulating retrotransposons silencing.”

Reply: Thank you very much for this positive assessment.

“Below are some detailed comments:

1. DNMT3C flanking sequence preference has been proved to affect cellular CpG methylation pattern in mouse ES cells. One question is whether in the absence of DNMT3C, these retrotransposons can still be methylated? Or the function can be rescued by other DNMT3s?”

Reply: It has been shown that the silencing of young retrotransposons in mice critically depends on DNMT3C (Barau et al. 2016, Jain et al. 2017). More specifically, methylation of the promoter regions of these elements was disrupted by a DNMT3C KO, but not by a DNMT3A KO (Dura et al. 2022). This information has been added to the introduction of our manuscript.

“2. In Figure 3C and D, C543 and V547 mutants are used to study the methylation rates on two designed substrates with TA and CC respectively. How about other sequence combination for the substrates? This can be tried to obtain a more credible conclusion.”

Reply: The activities of DNMT3C, DNMT3B and DNMT3C C543N/V547A have been experimentally investigated in all flanking sequence contexts and two analyses of the results are shown in Fig. 2 (focusing on individual flanking positions) and Fig. 4 (focusing on NNCGNN sequences). Most importantly, in both figures it is apparent that the C543N/V547A double mutation converts the DNMT3C-like flanking sequence preference pattern into a pattern very similar to DNMT3B. This observation indicates that the molecular interaction responsible for the change in flanking sequence preferences of DNMT3C when compared to its evolutionary relative DNMT3B are mediated by these two amino acid residues.

“3. Another question is for this specific binding preference, is it due to DNMT3C alone or the interaction with other members of the DNMT family, such as DNMT3A and DNMT3B?”

Reply: This is an interesting question. We have analyzed DNMT3C alone, as indicated in our paper. Currently, there is no information available regarding the potential complex formation of DNMT3C with DNMT3A, DNMT3L or DNMT3B (the latter is unlikely, given the distinct expression profiles of DNMT3C and DNMT3B). Moreover, methylation data of Dura et al. (2022) in DNMT3A and DNMT3C KO at the promoters of young retrotransposons revealed maintenance of methylation in DNMT3A KO but loss of methylation in DNMT3C KO, suggesting that DNMT3C does not form a complex with DNMT3A when introducing this methylation. This information has been added to the introduction of our manuscript.

“4. It is mentioned that the binding of DNMT3C to retrotransposons enhances their methylation and silencing during early development. so is the expression pattern of DNMT3C having any developmental stage or tissue specificity? It would be beneficial to investigate the differences in expression dynamics between DNMT3C and other DNMT3s in different developmental stages.”

Reply: This information is available in the literature. In fact, the expression profile of DNMT3C and DNMT3A2 is roughly overlapping in the male germ line. Still DNMT3C is required for efficient methylation of specific types of repeat elements. DNMT3B is not expressed in germ cells. We have added this background information together with the related references to the introduction of our paper.

“5. Are there any DNMT3C ChIP-seq previously been performed? Can authors make use of ChIP-seq data to further confirm whether binding specificity of DNMT3C is also correlated with the specified flanking sequences in the genome?”

Reply: Unfortunately, no DNMT3C ChIP-seq data are available. Moreover, available ChIP-seq profiles of DNMT3A and DNMT3B are not reflecting the DNA binding preferences of their catalytic domains, but they are highly dominated by chromatin binding through the PWWP and UDR domains (see for example: Baubec et al. 2015 Nature 520: 243-247, Weinberg et al. 2019 Nature 573: 281-286, Weinberg et al. 2021 Nat Genet 53: 794-800). Hence similar studies with DNMT3C are unlikely to provide data relevant for our study, which deals with the properties of the catalytic domain of DNMT3C.

“Overall, I think this is a nice investigation on the sequence preference for DNMT3C, and the authors also extend the finding to further hypothesize how this differed flanking sequence preference affects specific retrotransposon silencing. It provides novel insights for future investigation on the sequence-specific silencing of retrotransposons during development. It also promotes us to look into the structural basis of different DNMT3 enzymes for their sequence preferences. Therefore, I think this paper has demonstrated value for the current research field.”

Reply: Once more thank you very much for this positive assessment. We hope that your helpful comments have been answered and addressed convincingly in the revision of our work.

REVIEWERS' COMMENTS:

Reviewer #1 (Remarks to the Author):

The authors have addressed the concerns and improved the manuscript. I have no further comments.

Reviewer #2 (Remarks to the Author):

The authors have addressed my questions, and I have no further comments for the revised version of the manuscript.